# Compassionate Engagement and Action in the Education for Health Care Professions: A Cross-Sectional Study at an Ecuadorian University

**DOI:** 10.3390/ijerph17155425

**Published:** 2020-07-28

**Authors:** Viviana Davalos-Batallas, Ana-Magdalena Vargas-Martínez, Patricia Bonilla-Sierra, Fatima Leon-Larios, Maria-de-las-Mercedes Lomas-Campos, Silvia-Libertad Vaca-Gallegos, Rocio de Diego-Cordero

**Affiliations:** 1Health Sciences Department, Private Technical University of Loja, Loja 110107, Ecuador; vddavalos@utpl.edu.ec (V.D.-B.); pbonilla65@utpl.edu.ec (P.B.-S.); slvaca@utpl.edu.ec (S.-L.V.-G.); 2Nursing Department, School of Nursing, Physiotherapy and Podiatry, University of Seville, 41009 Seville, Spain; mlomas@us.es (M.-d.-l.-M.L.-C.); rdediego2@us.es (R.d.D.-C.)

**Keywords:** compassion, compassionate universities, cross-sectional study, education, health professions, palliative care, self-compassion

## Abstract

*Objective:* This study aimed at exploring the compassion attitudes and needs for awareness and training related to a compassionate approach for Medicine, Nursing, and Psychology students, as well as for the academic and administration personnel from the Universidad Técnica Particular de Loja (UTPL, Ecuador) Health Sciences area. *Methods*: A cross-sectional observational study, based on a self-administered questionnaire through a sample of 788 UPTL students. STROBE guidelines were followed and applied. *Results:* A positive correlation was found between life engagement and compassion for others, from others, and self-compassion. The Nursing students were those who reported having previous experiences of contact with people with an advanced disease or in an end-of-life situation and having received some type of training compared to Medicine and Psychology students and lecturers (faculty members). Differences were found on the “self-compassion” and “compassion for others” subscales, noting a higher level of compassion among Psychology students. *Conclusions*: To implement the philosophy of compassionate universities it is necessary to design trainings that include the students, the faculty members, and the administrative staff, centered on sensitization and training about assistance, care, and accompaniment at the end of life, as well as cultivating compassion in the workplace.

## 1. Introduction

During the last decades, we have been witnessing a progressive aging of the population worldwide. By 2050, one out of six people (16%) in the world will be over 65 years old, an increase from one out of 11 (9%) in 2019. Latin America regions are included among those countries where the population aged 65 years old or over will probably double. In Europe, the number of people aged 80 years old or over is projected to triple [1]; therefore, health promotion approaches focused on public policies and healthy environments will be necessary to improve quality of life [2,3,4]. Aging may be accompanied by an increase in frail older adults as a consequence of unfavorable conditions such as great social inequality, poor access to health services, low coverage of health policies, and a decrease in the informal supply of care services. Identifying the population groups that are most vulnerable and providing advice on suitable interventions to reduce existing inequalities are two main objectives [5,6].

This population aging is the consequence of a longer life expectancy and will bring with it a notable increase in chronic and degenerative diseases and a higher level of dependence. As a result, there will also be a greater need for palliative care and new approaches to care must be considered because, in some cases, palliative care is oriented to health development and it sometimes does not fully cover social needs [6,7,8,9]. As a result, there is an extreme need for palliative care, integrated into all the specializations; many times, this is not possible due to lack of time, very specialized and fragmented Medicine, lack of recognition of end-of-life situations by the professionals, and limited communication skills as regards bad news, among others [10]. In these circumstances of chronicity, dependence, and end-of-life care, patients’ needs cover multiple domains of symptom control, social environment, psychological and emotional distress, and spiritual care [9,10,11]. Therefore, it is necessary to develop care models centered on the patient and on the fight against inequality while also providing holistic care, including the spiritual approach [12]. In this framework, the compassionate communities movement has emerged to respond to the needs of these patients and to a greater humanization of health care, building community networks [7,11,13,14,15,16,17].

Compassion can be defined as the sensitivity shown to understand another person’s suffering, combined with a willingness to help and to promote the well-being of that person, in order to find a solution to their situation [18]. Compassionate care includes ethical, professional, effective communication, human, and spiritual/religious dimensions. Thus, to include educational interventions in health organizations and health care staff training programs oriented to these dimensions is completely necessary to improve compassionate care and the quality of palliative care treatments [19,20]. Compassionate communities offer the possibility of reducing inequality, promoting awareness, education, and community commitment to provide support and end-of-life care [12,15].

Compassion is considered an essential component of quality care. Patients expect the health care staff to treat them with compassion, but this does not always happen due to organizational factors in the workplace and to factors related to lack of competence in the knowledge, attitudes, and skills necessary for compassion by the health personnel. However, compassion can be cultivated and there are models to optimize it at a personal, relational, and system level [21], among which are initiatives in universities to teach the values of care by practicing compassion [15].

Similarly to what happens in Latin America, in Ecuador the population continues to increase and age steadily [22], and palliative care is a relatively new concept for the Ecuadorian society. People generally die at their homes, in hospitals, or in nursing homes. In 2012, the first Ecuadorian hospice was created and research began on end-of-life care, but strategies are needed to intensify the training and competences of medical, nursing, and other professionals in palliative care [23].

Previous studies have questioned if the health professionals are trained for caring patients compassionately. The patients often expect this type of care; however, the health professionals are not always prepared to offer it [24]. In a previous mixed-method systematic review, researchers showed that the impact of compassionate care educational programs on Nursing leaders, students, and educators, among others, was positive; the main barriers being identified in relation to lack of available resources, lack of time, and lack of support, showing the importance of the workplace culture and team relationships [25,26,27], of support, of staffing, and of resources.

The importance of including educational contents related to compassionate care in the studies related to Health Sciences has been widely recommended in the literature [28,29,30]. Contents related to compassionate attitudes have helped to develop a reflexive practice, deal with the clinical challenges, and gain confidence in the health professionals, such as nurses [25]. In the case of the Medical professionals, recent studies have showed that the physicians practicing an empathetic and compassionate attitude is one of the most important aspects of the therapeutic relationship, its effects being measurable through specific neurobiological mechanisms that demonstrate the positive effect of compassionate medical care [31]. Among the factors most valued by students and professionals for the development of compassionate care are acting with warmth and empathy, providing patient-centered care, and treating others as you would like to be treated [32]. Compassionate care should be present in any act of therapeutic interrelation, but even more in those related to processes that entail a high load of associated suffering like chronic or disabling process, or in special circumstances such as end-of-life care. Additionally, compassion can be understood as yet another indicator of a good care practice, contributing an ethical and spiritual dimension to integral or holistic care, reason why it should be practiced by all health professionals. In the absence of previous data, we have carried out this work. The general objective is to establish a baseline from which to develop strategies and initiatives that promote the feeling of the Universidad Técnica Particular of Loja (Private Technical University of Loja, UTPL) as a compassionate university, the first of reference in Ecuador.

The specific objectives are the following: to explore the compassion attitudes and needs for awareness and training related to a compassionate approach for Medicine, Nursing, and Psychology students, as well as for the faculty and administration personnel from the UTPL Health Sciences area.

## 2. Materials and Methods

### 2.1. Study Design and Settings

A cross-sectional and observational study conducted from June 2019 to January 2020 in the Private Technical University of Loja, Ecuador. The STrengthening the Reporting of OBservational studies in Epidemiology (STROBE) guidelines for cross-sectional studies were followed and applied in this study [33].

### 2.2. Sample Size, Sampling, and Study Variables

A convenience sample was recruited made up by the universe of enrolled students in the Medicine, Nursing, and Psychology courses of the UTPL. Participation was also asked from the professors teaching those courses, as well as from the administrative and service staff from the Health Sciences area of the University. The universe to be interviewed consisted of 894 students enrolled in the Medicine (455), Psychology (339), and Nursing (100) courses, 77 full- and part-time professors, and 27 administrative and service clerks.

This intervention is framed in the project entitled “Universidad compasiva, Universidad contigo” (The Compassionate University, the University with You), jointly started in 2018 by the UTPL and the New Health Foundation to work on humanization, dignity of the individuals, and compassion, not only from teaching but also from research.

### 2.3. Measures

As evaluation instrument, an ad hoc questionnaire was prepared structured in five blocks: (a) sociodemographic characteristics; (b) level of knowledge and sensitivities on care, accompaniment and assistance to people at the end of their lives; (c) training needs of the professors and students in relation to the care for individuals with an advanced disease and/or at the end of their lives; (d) valuation of compassion; and (e) valuation of the commitment to life. The first three blocks were defined based on previous literature, and the last two blocks were validated questionnaires.

For Section d, Gilbert’s validated questionnaire (2017) on self-compassion, compassion for others, and compassion from others based on the model of compassion competencies was used. As Gilbert defined, self-compassion is being sensitive to, and moved by one’s suffering, compassion for others is the compassion that we experience for others; compassion from others is referred to the experience of compassion from people around us, whether we feel they are supportive and have compassion competencies [34]. Each of these scales has two sections: (1) *Engagement*: questions to investigate the person’s motivation and involvement in suffering when experiencing it; (2) *Action:* questions on the way in which the individuals compassionately face the emotions, thoughts, and situations that anguish them, consisting in 5 items. Each of the items is answered by means of a Likert-type scale from 1 to 10, where 1 is “never” and 10 is “always”. The Cronbach´s alpha for the different subscales was from 0.72 to 0.94.

To assess the purpose in life, the Life Engagement Test was used [35]. The intent of the scale, consisting of 6 items, is to provide an index of purpose in life by assessing the extent to which a person considers their activities to be valuable and important. Each item is scored from 1 “strongly disagree” to 5 “strongly agree”. A higher index indicates higher optimism or purpose in life. The Cronbach’s alpha for the different samples analyzed in the validity study carried out by Scheier et al. [35] was from 0.72 to 0.87.

The original questionnaires prepared in English were translated into Spanish by a bilingual member of the research team. Subsequently, the corresponding back-translation into English was performed to guarantee accuracy and fidelity. The result was discussed by the research group and any inconsistency or unclear concept was agreed upon among subject matter experts in Medicine, Nursing and Psychology with a high level of English. The resulting questionnaire was piloted among students and professors prior to the beginning of the recruitment period to define the inconsistencies or difficulties identified. These were solved and the questionnaire was considered ready for its distribution. For data collection, a paper copy of the questionnaire was handed in to each participant for it to be self-administered during or after the classes. In the case of the workers, it was distributed during their working hours for them to complete it.

### 2.4. Statistical Analysis

The sociodemographic characteristics of the subjects and the general aspects about knowledge on palliative care and experiences related to it were described as means ± standard deviations, or n (%) of the total. Additionally, ANOVA, Pearson’s χ² and Fisher tests, depending on the type of variable and its answer’s frequency, were used to compare these characteristics by the students’ course (Medicine, Nursing, Psychology) and by affiliation with the University (student, faculty members, and administration staff). An analysis was conducted of the training needs of students and faculty members related to the care of individuals with an advanced disease and/or at the end of their lives by the students’, by affiliation with the University (student, faculty member), and by gender. For this, Pearson’s χ² was used when the variable was qualitative, and parametric tests (t-test or ANOVA) and non-parametric tests (Kruskal-Wallis and Wilcoxon) were used when the variable was numeric and depending on its normality. In relation to the Compassion subscales (self-compassion, compassion from others, compassion for others) and to the Life Engagement Test, an analysis of mean comparisons was carried out by the students’ course, by affiliation with the University (student, faculty members, administration staff), and by gender. In addition, Spearman rank-order correlation coefficients were calculated to explore the relationships between the three orientations of compassion, compassion focused, self-evaluative and emotion focused variables, due to non-normal distribution of different subscales. These coefficients were classified as follows: “high degree” if the coefficient value lies between ±0.50 and ±1, then it is said to be a strong correlation; “moderate degree” if the value lies between ±0.30 and ±0.49, then it is said to be a medium correlation; and “low degree” when the value lies between 0 and ±0.29, then it is said to be a small correlation.

The associations between willingness to dedicate themselves professionally to Palliative Care and the compassion level measured through Gilbert’s scale controlled by different covariates (age, gender, affiliation with the University, feeling trained to accompany a person in the stages of their disease, optimism level measured through the Life Engagement Test) were studied using logistic regression analysis.

Some variables were centered using medians, such as age, because a value of 0 was not included in these variables for the sample studied. Data was entered into the R software (R-3.6.3 version, Free Software Foundation’s GNU General Public License: https://www.r-project.org/about.html) for statistical analysis. A *p* value < 0.05 was accepted as statistically significant.

### 2.5. Ethical Considerations

Participation was asked from all the individuals who met the aforementioned criteria. Once they accepted, their written consent to participate was asked after having informed them the study objectives. Anonymity and data confidentiality were guaranteed to the participants, with data being employed solely for this research. All the data derived from this study were treated following the recommendations set forth in the Declaration of Helsinki, and its subsequent amendments. This study was approved by the commission on research studies by peer-review of the Private Technical University of Loja, Ecuador (PROY_INV_CCSAL_2018_2367) prior to the beginning of recruitment.

## 3. Results

### 3.1. Descriptive Characteristics of the Sample

Initially, a total of 809 subjects comprised the sample, representing a response rate of 81.06%. Of these, 21 subjects did not provide data in relation to their affiliation with the University so they were deleted, resulting in a total of 788 subjects. The training needs related to the care of people with an advanced disease and/or at the end of their lives corresponding to section “c” of the complete questionnaire that was administered only was addressed to students and faculty members, so these data are available for a sample comprised of 536 subjects (459 students and 77 lecturers). A total of 225 Psychology students dropped out, meaning that they did not fill this section of the questionnaire.

In relation to the sociodemographic characteristics, the mean age was 26.84 (SD 9.40) years old with a range between 18 and 70 years old, and 75% of the subjects were under 32 years old. Statistically significant differences related to age were observed among the Medicine, Nursing, and Psychology students, as well as among the students in general, the faculty members, and the administration staff. The proportion of women and men showed a statistically significant difference in all the subgroups (students, faculty members, administration staff), with a higher proportion of women within each subgroup, highlighting higher percentages among the Nursing students and administration staff subgroups. In the subgroup of students, the household structure consisted mostly of a single parent home with children, whereas in the faculty members and administration staff subgroups, the household structure consisted mostly of a home made up by a couple with or without children (Table 1).

Concerning general aspects of palliative care, the statistically significant difference found in relation to knowledge about them among students of the different university courses stands out, highlighting a higher percentage of students who claimed to know what Palliative Care is in the Medicine and Nursing courses. Among the subgroups explored, the Nursing students were those who reported having previous experiences of contact with people with an advanced disease or in an end-of-life situation in a greater proportion (85.2%). In relation to the conceptualization of palliative care, the students largely associated it with decreased suffering, while among the faculty members and administration staff, palliative care was associated with quality of life in a higher proportion (Table 1).

### 3.2. Training Needs of Lecturers and Students Related to the Care of Persons with an Advanced Disease and/or at the End of Their Lives

Regarding the training received in palliative care as well as feeling personally capable of accompanying a person in the stages of their disease, a higher percentage of Nursing students reported having received some type of training compared to Medicine and Psychology students and lecturers (faculty members). However, a higher proportion of Psychology students reported that they would like to dedicate themselves professionally to Palliative Care. 

No statistically significant differences were found in relation to the priority level given to a series of competences to be developed in training related to palliative care by affiliation with the University (student or lecturer). This means that both students and lecturers reported similar priority levels to the different palliative care training competencies asked. Nevertheless, some differences were found among students of different courses. In the total sample, the competences considered with the highest priority were “meeting the social needs of patients” and “caring with humanity, dignity, and compassion”, although the first competence mentioned was not one of the most highly prioritized competences among Medicine and Nursing students. Concerning the training received during the period at the university, the aspects of care mentioned in higher proportion in the total sample were those related to “death and grief”, “psychological aspects, information and communication at the end of life” and “needs of people with an advanced disease and/or at the end of their lives”. Related to the degree of interest in training these competences, a higher interest was observed among the students in general in comparison with the lecturers (Table 2). By gender, although a higher percentage of men reported having received training in palliative care than women, this last group stated in a greater proportion that they would like to dedicate themselves to palliative care professionally, this difference being statistically significant among the students. In relation to the degree of interest in training different competences to be developed related to palliative care among the students, women reported higher scores in general, reaching statistical significance (Table 3).

### 3.3. Gilbert’s Scale (2017): Self-Compassion, Compassion for Others, Compassion from Others; and Life Engagement Test

By affiliation with the University (student, lecturer, administration staff), statistically significant differences were not found in the scores of the different subscales that make up the Gilbert’s compassion scale used. However, comparing students from different courses, differences were found on the self-compassion and compassion for others subscales, noting a higher level of compassion among Psychology students (Table 4). By gender, in the “compassion for others” subscale, specifically those items that measure how the person compassionately copes with distressing emotions, thoughts, and situations (“Actions”), a statistically significant difference was found between women and men, with women scoring higher compassion levels (Table 5).

When assessing the purpose in life through the Life Engagement Test, differences were found both by affiliation with the University (*p* = 0.003) and by the student’s University course (*p* = 0.000) (Table 4), and by gender among the students (0.025) (Table 5). The subgroups that reported a higher level of optimism in life were lecturers, Nursing students, and female students.

All the correlations between these subscales were statistically significant and positive. For each specific approach (for oneself, for others, from others), the correlations between the components of engagement and action are high (r = 0.57 to 0.74). However, the correlations between different subscales for compassion are generally moderate except for engagement and compassionate actions for others with self-compassion actions, being 0.64 and 0.59, respectively (Table 6).

These data suggest that, while there are associations between different compassion orientations, they are only moderately associated. From this, it could be concluded that some people may be high in one type of compassion (for example, from others) but moderate in another (for example, for others) and vice-versa.

Since a validation of the Gilbert scale in the Ecuadorian population was not found in the literature, a reliability analysis was carried out in the sample of students, finding a Cronbach’s alpha of 0.85 for the global scale, varying between 0.81 and 0.83 for the different subscales that compose it.

When analyzing the correlations between the compassion subscales and the Life Engagement Test, all the correlations were statistically significant and positive (Table 7). However, these correlations between the components of engagement and action of the different compassion subscales and the Life Engagement Test were mainly lowly associated, except for the component action of the “self-compassion” and “compassion for others” subscales, which were moderately associated.

### 3.4. Associations between Willingness to Dedicate Themselves Professionally to Palliative Care and Compassion Level

Initially, when analyzing the association between the willingness to dedicate themselves professionally to palliative care and the compassion level through a univariate logistic regression model in students and lecturers, women and those who reported feeling trained to accompany a person in the stages of their disease would have greater willingness to dedicate themselves professionally to palliative care. Similarly, a higher score in the different compassion subscales and in the Life Engagement Test was found to report willingness to dedicate themselves professionally to palliative care. In the multivariate logistic regression model, being a woman as well as an increase in age and in compassion for others were found to report willingness to dedicate professionally to PC, whereas lecturers in comparison to students were found to present lower willingness to dedicate themselves professionally to palliative care (Table 8).

Similar findings were verified in the logistic regression models carried out in the sample of students. In the univariate logistic regression model, Nursing and Psychology students were found to report higher willingness to dedicate to palliative care than Medicine students. This finding was also found in the multivariate model, although only for Psychology students. A difference was found in the comparison with the model carried out in lecturers and students in relation to age, whose association with the willingness to dedicate to palliative care was not found (Table 9).

## 4. Discussion

The number of research studies is ever increasing that resort to the self-reporting measures of compassion [34,36]. The aim of this study was to know the level of self-declared compassion among students, faculty members, and administrative and service staff. By exploring these compassionate attitudes, the objective is to identify the needs for sensitization and training in order to address compassion at the end of life in the students and workers from an Ecuadorian university, Universidad Técnica Particular de Loja (Private Technical University of Loja, UTPL), which aspires to be recognized as a compassionate university.

Compassion is a component in health care that involves awareness of and participation in the suffering of another person, reducing the suffering observed [18,37]. It is crucial to train the next generations of health students in the relief of patients’ suffering [15]. The participants in this study defined compassionate care as that which seeks to relieve suffering in others and improve their quality of life, a definition that is in consonance with that provided by other authors [34]. Thus, in relation to compassion care, it was found that the students had broad knowledge about the meaning of compassion, as stated by Bickford et al. [38].

In this study, Gilbert’s validated scale on compassion competences [34] was used, since the objective was to explore sensitivity and commitment with suffering, as well the actions of people to relieve and prevent suffering. On the other hand, we found the approach of exploring compassion from its different aspects to be interesting: self-compassion, compassion for others, and compassion from others, thus allowing evaluating the interactions from different perspectives. Likewise, the purpose in life was measured by the Life Engagement Test. This instrument allows studying behaviors and health outcomes, having been proved that it correlates with psychosocial factors, such as dispositional optimism and emotional expression style [35]. The results derived from this research show how compassion for others, from others, and self-compassion are positively correlated with life engagement. In this way, we can assert that those people who show greater life engagement also present more compassionate behaviors, according to Asano et al., who showed medium correlation with satisfaction with life [36]. However, it is not only related to compassion for others, but also to compassion from others, and to self-compassion.

With this research, we set out to determine if students of areas related to the Health Sciences, as well as the professors and the administrative and service staff working in the UTPL, possess this compassionate competence. We were interested in knowing the factors associated to a more compassionate behavior in the scope of Palliative Care in the future health professionals, and in identifying gaps that could be bridged by means of specific training. Among these associated factors, creating a culture of compassion that starts from the first periods of schooling was highlighted across several studies. In relation to the gaps, in a previous study with nurses, lack of communication skills was identified as the main gaps [39]. Lack of information at all levels (undergraduate, graduate) and in all affiliations (students and professors] were also identified as an element to be improved [40,41]. However, another study pointed out that the knowledge about Palliative Care among undergraduate Nursing students expands with the student’s age and academic year of study, and that the attitudes towards end-of-life care improve as the students progress in their studies [42], which reinforces the aforementioned idea of incorporating training in palliative care at early levels.

Regarding the training received on Palliative Care, as well as feeling personally capable of accompanying a person in the stages of their disease, a higher percentage of Nursing students reported having received some type of training compared to Medicine and Psychology students, thus faculty members and administration staff. However, a higher proportion of Psychology students reported that they would like to dedicate themselves professionally to Palliative Care. In our study, Nursing students reported previous experiences in Palliative Care compared to other groups of students, associating this care with decreased suffering. These findings are interesting to organize trainings in the future, taking into account these differences between groups of students depending on the career that they are involved in. In this regard, other studies have pointed out that Nursing students showed positive attitudes towards end-of-life care but less positive attitudes with the care of a dying person and their imminent death [42].

Previous studies conducted with students and professors observed that there were difficulties in the development of compassion during health care training, sometimes due to the level of compassion of the professors [24]. This, together with previous experiences like the Stanford Compassionate University Project or the University of Worcester, which was the first British university to sign the Charter for Compassion, drove us to include not only students in our study but also professors and administrative staff, noting that they showed high levels of compassion. This can better contribute to developing strategies for encouraging compassion, since the University will be compassionate if its faculty and administrative staff show high levels of compassion, which will contribute to the students training in this setting and, at the same time, develop compassionate attitudes.

Whereas the results of Gilbert et al. [34] did not show differences between men and women in measuring self-compassion and compassion from others, but did found them in compassion for others, our results were in line with this and showed a higher level of compassion for others, especially in women [34]. This is important to highlight in our study, with the identification of a pronounced influence of gender, since more women were found who are interested in professional Palliative Care and in acquiring competences in the practice.

Compassion is a social process which can be influenced by people’s previous experiences. This is in consonance with our findings, which evidence that the individuals with previous experience in providing Palliative Care to people they know show higher levels of compassion. These results are in line with the research study conducted by Adamson and Dewar, who identified how previous experiences and the very own values of the students influence on the way they provide person-centered compassionate care. Hence the importance of sensitizing the students on compassionate care from the early years of training [41].

Regarding the training on compassionate care, the recent literature evidences a decreasing trend to include compassion in the training of health professionals [43,44,45] despite the stated benefits both for patients and for professionals [28]. Providing compassionate care is something to be learned and trained [46,47,48] and its inclusion in the curricula is the first step for its implementation in the clinical practice [19,20]. Compassion is the key value from the perspective of the Humanistic Sciences, addressing the non-physical aspects of palliative care. Exercising the Humanities makes the clinical setting value the history of suffering, meaning, and the need for the patient’s empathy and compassion [49]. This is the reason why we believe that it of utmost importance to cultivate a compassionate attitude in the Health Sciences student and to recover the true sense of service which humanizes the professional [11,13,28,29,30,31,32].

Nevertheless, a number of previous studies have showed the difficulty of developing compassion in the training of health professionals [50] who, despite considering compassion as a very important element for their training, do not feel competent enough to apply compassionate care in the practice [51]. This is the reason why it is so important that the students be trained if they consider this need as a priority (as in the case of the Nursing students) [25] and if their professors are competent enough to put it into practice [24], and future research should be addressed for this topic.

The limitations we found in this study are the ones inherent to the instrument, since it is possible that the respondent does not really grasp the essence of what is being conveyed. Compassion is not related to other emotional aspects which are intimately linked, like empathy. Both empathy and compassion are essential in caring for the patients. Additionally, empathy is required to develop compassion [52].

This is a cross-sectional study, so its results cannot be attributed to causality. It would be interesting to conduct experimental studies that measure the change in compassion and assess this follow-up during the students’ years of study. On the other hand, as this study was conducted in a single university, it is difficult to generalize its results to other universities or curricula.

This study supposes a starting point to integrate compassionate care in the Ecuadorian universities and as a component of the curricula for the courses related to Health Sciences. It allows knowing the interests of the University members on compassionate care and what they understand by this type of care, as well as associating the influencing factors. From this, trainings can be designed aimed at the students, the faculty members, and the administrative staff, thus contributing to the implementation of the philosophy of compassionate universities setting in motion actions of sensitization and training on end-of-life assistance, care, and accompaniment, as well as cultivating compassion in the workplace.

## 5. Conclusions

There is a positive predisposition towards compassionate care, understanding it as that which aims to relieve suffering in others and to improve their quality of life. A gender influence is observed in the interest awakened by compassionate care, especially among women, which is also marked by previous experiences caring for individuals with palliative care needs. There is an evident interest in compassionate care in the courses of the Health Sciences area, but is necessary to improve the training and sensitization in undergraduate students, as well as to evaluate the impact of these trainings on the compassion competences.

It is necessary to implement more measures that contribute to the organizational culture both at the professional practice level in the clinical settings and in the teaching of compassion in the educational spaces. These factors would have to be taken into account for the development of strategies that contribute to the creation of compassionate spaces in the University. This study supposes a starting point for this University, which aims at developing and integrating contents and training on compassion for its students in the Health Sciences area, such as creation of a web, volunteer actions, death cafes, and thank you notes in the trees of the University, among other activities. It has already been verified that compassionate care is not trained in University, although its usefulness has been proved, reason why this study can contribute to this. Hence the need to include and potentiate the philosophy of compassionate care in the clinical settings of learning of the Health Sciences students.

## Figures and Tables

**Table 1 ijerph-17-05425-t001:** Characteristics of the subjects and general aspects about Palliative Care (n = 788).

Variables	Total Sample n = 788	Students n = 684			*p*-Value ^a^	Faculty Members n = 77	Administration Staff n = 27	*p*-Value ^b^
		Medicine n = 277	Nursing n = 27	Psychology n = 380				
**Sociodemographic characteristics**								
**Gender**					0.000 ***			0.040 **
Female	577 (73.22)	176 (63.5)	24 (88.9)	300 (78.9)		52 (67.5)	25 (92.6)	
Male	211 (26.78)	101 (36.5)	3 (11.1)	80 (21.1)		25 (32.5)	2 (7.4)	
**Age**	26.84 (9.40)	21.46 (3.90)	21.74 (1.93)	27.62 (8.60)	0.000 ***	41.00 (10.28)	36.37 (9.94)	0.002 **
**Household structure**					0.004 **			0.000 ***
Home made up by a couple with/without children	196 (27.41)	41 (14.8)	3 (11.1)	115 (30.3)		43 (55.9)	14 (51.8)	
Single parent home with children	404 (51.27)	183 (66.1)	19 (70.4)	183 (48.2)		14 (18.2)	5 (18.5)	
Living alone	66 (8.38)	23 (8.3)	4 (14.8)	30 (7.9)		6 (7.8)	3 (11.1)	
Other	102 (12.95)	30 (10.8)	1 (3.7)	52 (13.7)		14 (18.2)	5 (18.5)	
**Palliative Care**								
**Do you know what Palliative Care is?** [yes]	611 (78.03)	241 (87.3)	23 (85.2)	263 (69.8)	0.000 **	65 (85.5)	19 (70.4)	0.171
**Who do you think are addressed?**					0.091 *			0.435
The entire population (adult and pediatric population) in a situation of chronic, advanced disease and/or at the end of their lives.	701 (88.96)	256 (92.4)	23 (85.2)	332 (85.2)		68 (88.3)	22 (81.5)	
Only the adult population or only the pediatric population in (…)	87 (11.04)	21 (7.6)	4 (14.8)	48 (12.6)		9 (11.7)	5 (18.5)	
**Previous experiences of contact with people with an advanced disease and/or in an end-of-life situation** [yes]	513 (65.1)	198 (71.5)	23 (85.2)	218 (57.4)	0.000 ***	60 (77.9)	14 (51.9)	0.019 **
**If previous experiences; what relationship did they have?**					0.000 ***			0.000 ***
Friend/Classmate/Co-worker	84 (17.04)	29 (14.65)	6 (26.09)	40 (18.35)		6 (10)	3 (21.43)	
Family	215 (43.61)	80 (81.63)	7 (30.43)	97 (44.50)		29 (48.33)	5 (35.71)	
Patient	58 (11.76)	27 (13.64)	3 (13.04)	11 (5.05)		17 (28.33)	0 (0)	
Other	136 (27.59)	52 (26.26)	6 (26.09)	65 (29.82)		8 (13.33)	5 (35.71)	
**Care of a family member/friend/peer who has been in an advanced disease process and/or end-of-life situation** [yes]	267 (33.88)	92 (33.2)	15 (55.6)	116 (30.5)	0.026 **	38 (49.4)	6 (22.2)	0.006 **
**Willingness to care for a person who is NOT from your family or from your closest environment** [yes]	676 (85.79)	255 (92.1)	27 (100)	321 (84.5)	0.002 **	55 (71.4)	18 (66.7)	0.000 ***
**Discussing death with your family, friends, coworkers, classmates**					0.207			0.224
Rarely or never	262 (33.25)	81 (29.2)	9 (33.3)	140 (36.8)		19 (24.7)	13 (48.1)	
Sometimes	264 (33.50)	106 (38.3)	8 (29.6)	116 (30.5)		27 (35.1)	7 (25.9)	
Always	262 (33.25)	90 (32.5)	10 (37.0)	124 (32.6)		31 (40.3)	7 (25.9)	
**What concept do you associate Palliative Care with?** *(Multiple choice response)*								
Death	64 (8.12)	22 (7.94)	2 (7.41)	31 (3.93)	0.000 ***	7 (9.09)	2 (7.41)	0.002 **
Agony and end of life	250 (31.73)	99 (35.74)	5 (18.52)	125 (32.89)	0.000 ***	17 (22.08)	4 (14.81)	0.002 **
Decreased suffering	512 (64.97)	206 (74.37)	17 (62.96)	233 (61.32)	0.000 ***	41 (53.25)	15 (55.56)	0.002 **
Quality of life	451 (57.23)	183 (66.06)	15 (55.56)	180 (47.37)	0.000 ***	57 (74.03)	16 (59.26)	0.002 **
The privilege of caring and being cared for	271 (34.39)	106 (38.27)	11 (40.74)	125 (32.89)	0.000 ***	19 (24.68)	10 (37.04)	0.002 **

*Note:* We show the mean values and standard deviations in brackets when the variable is numeric. We show the frequency and percentage in brackets when the variable is categorical. ***, **, and * represent statistically significant differences at 1%, 5%, and 10% between values of variables by students’ course ^a^ and by affiliation with the University ^b^ (students, faculty members, administration staff).

**Table 2 ijerph-17-05425-t002:** Training needs of lecturers and students related to the care of persons with an advanced disease and/or at the end of their lives (n = 536).

Variables	Total Sample n = 536	Students (n = 459)			*p*-Value ^a^	Faculty Members n = 77	*p*-Value ^b^
		Medicine n = 277	Nursing n = 27	Psychology n = 155			
**Training on Palliative Care [Yes]**	294 (54.85)	175 (63.2)	22 (81.5)	57 (63.2)	0.000 ***	40 (51.9)	0.580
**Training on Palliative Care (hours)**					0.001 **		0.655
20–40	184 (74.19)	116 (77.3)	11 (57.9)	29 (70.7)		28 (73.7)	
41–140	46 (18.55)	29 (19.3)	7 (36.8)	4 (9.8)		6 (15.8)	
More than 140	18 (7.26)	5 (3.3)	1 (5.3)	8 (19.5)		4 (10.5)	
**Do you feel personally able to accompany a person in the stages of their disease? [Yes]**	165 (30.78)	75 (27.1)	12 (44.4)	56 (36.1)	0.046 **	22 (28.6)	0.650
**Would you like to dedicate yourself professionally to Palliative Care? [Yes]**	251 (46.83)	109 (39.4)	16 (59.3)	95 (61.3)	0.000 ***	31 (40.3)	0.212
**Training received during the period at the University on the following aspects of care, accompaniment, and care for people with an advanced disease and/or at the end of their lives [Yes]**							
General concepts about what Palliative Care is.	314 (58.58)	209 (75.5)	21 (77.8)	49 (31.6)	0.000 ***	35 (45.5)	0.012 **
Profiles of the professionals involved in end-of-life care.	218 (40.67)	145 (52.3)	15 (55.6)	36 (23.2)	0.000 ***	22 (28.6)	0.020 **
Oncological Palliative Care.	144 (26.87)	93 (33.6)	19 (70.4)	22 (14.2)	0.000 ***	10 (13)	0.003 **
Non-oncological Palliative Care.	170 (31.72)	113 (40.8)	16 (59.3)	20 (12.9)	0.000 ***	21 (27.3)	0.365
Pediatric Palliative Care.	77 (14.37)	49 (17.7)	9 (33.3)	13 (8.4)	0.001 **	6 (7.8)	0.076 *
Needs of people with an advanced disease and/or at the end of their lives.	246 (45.9)	151 (54.5)	20 (74.1)	49 (31.6)	0.000 ***	26 (33.8)	0.021 **
Treatment of physical symptoms and other clinical problems at the end of life.	186 (34.7)	110 (39.7)	20 (74.1)	35 (22.6)	0.000 ***	21 (27.3)	0.139
Nursing care.	109 (20.34)	60 (21.7)	22 (81.5)	14 (9)	0.000 ***	13 (16.9)	0.416
Emergencies at the end of life.	120 (22.39)	65 (23.5)	12 (44.4)	26 (16.8)	0.005 **	17 (22.1)	0.944
Last days situation.	154 (28.73)	89 (32.1)	14 (51.9)	38 (24.5)	0.013 **	13 (16.9)	0.013 **
Death and grief.	286 (53.36)	144 (52)	16 (59.3)	87 (56.1)	0.598	39 (50.6)	0.607
Psychological aspects, information and communication at the end of life.	260 (48.51)	132 (47.7)	11 (40.7)	81 (52.3)	0.452	36 (46.8)	0.739
Social aspects at the end of life.	221 (41.23)	111 (40.1)	12 (44.4)	69 (44.5)	0.642	29 (37.7)	0.492
Cultural and spiritual aspects at the end of life.	217 (40.49)	117 (42.2)	14 (51.9)	58 (37.4)	0.316	28 (36.4)	0.426
Community aspects and the care network.	159 (29.66)	95 (34.3)	9 (33.3)	44 (28.4)	0.448	11 (14.3)	0.001 **
Tools for social awareness and development of actions from a community approach.	165 (30.78)	90 (32.5)	9 (33.3)	48 (31)	0.938	18 (23.4)	0.128
Palliative Care Volunteer Programs.	123 (22.95)	90 (32.5)	10 (37)	17 (11)	0.000 ***	6 (7.8)	0.001 **
Networking. Integrated and person-centered end-of-life care.	113 (21.08)	78 (28.2)	9 (33.3)	19 (12.3)	0.000 ***	7 (9.1)	0.005 **
Tools to care for and accompany people at the end of their lives.	159 (29.66)	100 (36.1)	16 (59.3)	27 (17.4)	0.000 ***	16 (20.8)	0.065 *
Skills to practice compassion, emotional balance, empathy, and active listening in the professional and personal practice.	225 (41.98)	123 (44.4)	16 (59.3)	63 (40.6)	0.194	23 (29.9)	0.02 **
Research and evaluation in Palliative Care.	126 (23.51)	87 (31.4)	17 (63)	17 (11)	0.000 ***	5 (6.5)	0.000 ***
Management and organization of Palliative Care resources.	91 (16.98)	63 (22.7)	10 (37)	14 (9)	0.000 ***	4 (5.2)	0.003 **
Compassionate community and network management.	88 (16.42)	58 (20.9)	11 (40.7)	16 (10.3)	0.000 ***	3 (3.9)	0.001 **
Public policies for the development of Palliative Care.	78 (14.55)	51 (18.4)	11 (40.7)	13 (8.4)	0.000 ***	3 (3.9)	0.004 **
Rights of end-of-life care.	182 (33.96)	119 (43)	16 (59.3)	33 (21.3)	0.000 ***	14 (18.2)	0.002 **
**Degree of interest in training on the following aspects of care, accompaniment, and care for people with an advanced disease and/or at the end of their lives (Likert 1-3; 1: “Low” to 3: ”High”)**							
General concepts about what Palliative Care is.	2.73 (0.51)	2.78 (0.47)	2.70 (0.47)	2.77 (0.44)	0.515	2.47 (0.66)	0.000 ***
Profiles of the professionals involved in end-of-life care.	2.65 (0.57)	2.71 (0.51)	2.63 (0.56)	2.70 (0.54)	0.704	2.35 (0.72)	0.000 ***
Oncological Palliative Care.	2.74 (0.55)	2.81 (0.49)	2.85 (0.36)	2.73 (0.53)	0.123	2.47 (0.74)	0.000 ***
Non-oncological Palliative Care.	2.74 (0.54)	2.81 (0.45)	2.81 (0.48)	2.74 (0.51)	0.313	2.45 (0.79)	0.000 ***
Pediatric Palliative Care.	2.72 (0.56)	2.82 (0.45)	2.78 (0.42)	2.73 (0.55)	0.242	2.34 (0.80)	0.000 ***
Needs of people with an advanced disease and/or at the end of their lives.	2.75 (0.51)	2.81 (0.43)	2.81 (0.48)	2.78 (0.50)	0.923	2.47 (0.68)	0.000 ***
Treatment of physical symptoms and other clinical problems at the end of life.	2.77 (0.50)	2.84 (0.39)	2.85 (0.36)	2.79 (0.48)	0.616	2.47 (0.77)	0.000 ***
Nursing care.	2.54 (0.66)	2.59 (0.60)	2.81 (0.48)	2.63 (0.58)	0.111	2.06 (0.86)	0.000 ***
Emergencies at the end of life.	2.75 (0.52)	2.82 (0.41)	2.85 (0.36)	2.78 (0.47)	0.682	2.39 (0.80)	0.000 ***
Last days situation.	2.73 (0.52)	2.77 (0.46)	2.70 (0.54)	2.81 (0.42)	0.457	2.42 (0.75)	0.000 ***
Death and grief.	2.73 (0.51)	2.74 (0.49)	2.70 (0.47)	2.80 (0.45)	0.288	2.57 (0.68)	0.022 **
Psychological aspects, information and communication at the end of life.	2.74 (0.52)	2.76 (0.47)	2.70 (0.54)	2.81 (0.48)	0.186	2.53 (0.70)	0.001 **
Social aspects at the end of life.	2.69 (0.54)	2.70 (0.50)	2.63 (0.56)	2.78 (0.47)	0.098 *	2.47 (0.70)	0.001 **
Cultural and spiritual aspects at the end of life.	2.66 (0.57)	2.68 (0.53)	2.67 (0.55)	2.75 (0.50)	0.270	2.43 (0.73)	0.001 **
Community aspects and the care network.	2.63 (0.59)	2.68 (0.54)	2.63 (0.56)	2.69 (0.54)	0.817	2.34 (0.77)	0.000 ***
Tools for social awareness and development of actions from a community approach.	2.66 (0.58)	2.71 (0.53)	2.70 (0.47)	2.69 (0.54)	0.852	2.38 (0.74)	0.000 ***
Palliative Care Volunteer Programs.	2.69 (0.55)	2.76 (0.50)	2.70 (0.54)	2.74 (0.48)	0.622	2.35 (0.72)	0.000 ***
Networking. Integrated and person-centered end-of-life care.	2.64 (0.57)	2.70 (0.54)	2.70 (0.54)	2.66 (0.53)	0.609	2.35 (0.70)	0.000 ***
Tools to care for and accompany people at the end of their lives.	2.75 (0.49)	2.80 (0.43)	2.70 (0.54)	2.81 (0.41)	0.590	2.45 (0.72)	0.000 ***
Skills to practice compassion, emotional balance, empathy, and active listening in the professional and personal practice.	2.76 (0.50)	2.82 (0.42)	2.67 (0.55)	2.81 (0.46)	0.207	2.48 (0.70)	0.000 ***
Research and evaluation in Palliative Care.	2.71 (0.54)	2.77 (0.49)	2.74 (0.45)	2.74 (0.47)	0.520	2.40 (0.75)	0.000 ***
Management and organization of Palliative Care resources.	2.64 (0.59)	2.70 (0.52)	2.70 (0.54)	2.73 (0.51)	0.792	2.22 (0.77)	0.000 ***
Compassionate community and network management.	2.63 (0.58)	2.71 (0.53)	2.70 (0.47)	2.66 (0.54)	0.610	2.27 (0.75)	0.000 ***
Public policies for the development of Palliative Care.	2.59 (0.60)	2.66 (0.55)	2.67 (0.55)	2.61 (0.59)	0.693	2.31 (0.77)	0.000 ***
Rights of end-of-life care.	2.76 (0.49)	2.81 (0.44)	2.78 (0.51)	2.81 (0.41)	0.973	2.47 (0.70)	0.000 ***

*Note:* We show the mean values and standard deviations in brackets when the variable is numeric. We show the frequency and percentage in brackets when the variable is categorical. ***, **, and * represent statistically significant differences at 1%, 5%, and 10% between values of variables by course ^a^ (Medicine, Nursing, psychology) and by affiliation with the University ^b^ (students, faculty members).

**Table 3 ijerph-17-05425-t003:** Training needs of the students by gender. (n = 459).

Variables	Students (n = 459)		
	Women n = 322	Men n = 137	*p*-Value
**Training on Palliative Care [Yes]**	165 (51.2)	89 (65)	0.007 **
**Training on Palliative Care (hours)**			0.665
20–40	105 (76.1)	51 (70.8)	
41–140	25 (18.1)	15 (20.8)	
More than 140	8 (5.8)	6 (8.3)	
**Do you feel personally able to accompany a person in the stages of their disease? [Yes]**	98 (30.4)	45 (32.8)	0.610
**Would you like to dedicate yourself professionally to Palliative Care? [Yes]**	172 (53.4)	48 (35)	0.000 ***
**Priority level for this series of competences to be developed in training related to Palliative Care (Likert 1–5, “1: Low priority” to 5: ”High priority”)**			
Applying the basic components of Palliative Care in the environment where the patients and their families are.	4.21 (1.10)	4.24 (0.97)	0.809
Increasing physical well-being along the path of the patients’ disease.	4.30 (1.04)	4.17 (0.99)	0.049 **
Meeting the psychological needs of the patients.	4.43 (1.03)	4.33 (0.99)	0.061 *
Meeting the social needs of the patients.	4.14 (1.05)	4.12 (0.99)	0.665
Meeting the spiritual needs of the patients.	4.37 (1.04)	4.25 (0.97)	0.041 **
Responding to the needs of caregivers.	4.24 (1.06)	4.16 (0.94)	0.085 *
Responding to the challenges of clinical and ethical decision-making in Palliative Care.	4.27 (1.05)	4.25 (0.97)	0.457
Coordination and interdisciplinary teamwork in a comprehensive way.	4.33 (1.04)	4.26 (0.95)	0.109
Developing interpersonal and communication skills.	4.33 (1.08)	4.34 (0.96)	0.532
Practicing self-awareness and a commitment to continued professional development.	4.34 (1.04)	4.23 (1.03)	0.182
Caring with humanity, dignity, and compassion.	4.50 (1.06)	4.45 (0.94)	0.072 *
Process integrated by health, social, and community care.	4.41 (1.04)	4.35 (0.93)	0.107
Identifying and managing networks around the patients.	4.31 (1.03)	4.20 (0.99)	0.106
Updating knowledge through research and evidence-based Medicine.	4.36 (1.04)	4.39 (0.93)	0.844
**Training received during the period at the University on the following aspects of care, accompaniment, and care for people with an advanced disease and/or at the end of their lives [Yes]**			
General concepts about what Palliative Care is.	193 (59.9)	86 (62.8)	0.569
Profiles of the professionals involved in end-of-life care.	121 (37.6)	75 (54.7)	0.001 ***
Oncological Palliative Care.	89 (27.6)	45 (32.8)	0.262
Non-oncological Palliative Care.	89 (27.6)	60 (43.8)	0.001 ***
Pediatric Palliative Care.	49 (15.2)	22 (16.1)	0.820
Needs of people with an advanced disease and/or at the end of their lives.	149 (46.3)	71 (51.8)	0.276
Treatment of physical symptoms and other clinical problems at the end of life.	109 (33.9)	56 (40.9)	0.151
Nursing care.	64 (19.9)	32 (23.4)	0.401
Emergencies at the end of life.	69 (21.4)	34 (24.8)	0.426
Last days situation.	90 (28.0)	51 (37.2)	0.049 **
Death and grief.	173 (53.7)	74 (54.0)	0.955
Psychological aspects, information and communication at the end of life.	146 (45.3)	78 (56.9)	0.023 **
Social aspects at the end of life.	126 (39.1)	66 (48.2)	0.072 *
Cultural and spiritual aspects at the end of life.	118 (36.6)	71 (51.8)	0.002 ***
Community aspects and the care network.	91 (28.3)	57 (41.6)	0.005 **
Tools for social awareness and development of actions from a community approach.	91 (28.3)	56 (40.9)	0.008 **
Palliative Care Volunteer Programs.	67 (20.8)	50 (36.5)	0.000 ***
Networking. Integrated and person-centered end-of-life care.	59 (18.3)	47 (34.3)	0.000 ***
Tools to care for and accompany people at the end of their lives.	96 (29.8)	47 (34.3)	0.342
Skills to practice compassion, emotional balance, empathy, and active listening in the professional and personal practice.	136 (42.2)	66 (48.2)	0.241
Research and evaluation in Palliative Care.	71 (22.0)	50 (36.5)	0.001 **
Management and organization of Palliative Care resources.	49 (15.2)	38 (27.7)	0.002 **
Compassionate community and network management.	48 (14.9)	37 (27.0)	0.002 **
Public policies for the development of Palliative Care.	46 (14.3)	29 (21.2)	0.068 *
Rights of end-of-life care.	111 (34.5)	57 (41.6)	0.147
**Degree of interest in training on the following aspects of care, accompaniment, and care for people with an advanced disease and/or at the end of their lives (Likert 1–3; 1: “Low” to 3: ”High”)**			
General concepts about what Palliative Care is.	2.79 (0.44)	2.72 (0.51)	0.137
Profiles of the professionals involved in end-of-life care.	2.73 (0.50)	2.65 (0.55)	0.113
Oncological Palliative Care.	2.80 (0.48)	2.74 (0.55)	0.219
Non-oncological Palliative Care.	2.85 (0.45)	2.72 (0.50)	0.010 **
Pediatric Palliative Care.	2.82 (0.43)	2.69 (0.59)	0.026 **
Needs of people with an advanced disease and/or at the end of their lives.	2.84 (0.42)	2.70 (0.53)	0.002 **
Treatment of physical symptoms and other clinical problems at the end of life.	2.86 (0.39)	2.73 (0.49)	0.001 **
Nursing care.	2.67 (0.57)	2.50 (0.62)	0.003 **
Emergencies at the end of life.	2.85 (0.39)	2.72 (0.50)	0.003 **
Last days situation.	2.81 (0.42)	2.70 (0.52)	0.017 **
Death and grief.	2.79 (0.43)	2.69 (0.56)	0.167
Psychological aspects, information and communication at the end of life.	2.80 (0.47)	2.72 (0.50)	0.055 *
Social aspects at the end of life.	2.76 (0.47)	2.64 (0.54)	0.018 **
Cultural and spiritual aspects at the end of life.	2.72 (0.51)	2.64 (0.55)	0.099 *
Community aspects and the care network.	2.72 (0.51)	2.58 (0.59)	0.011 **
Tools for social awareness and development of actions from a community approach.	2.74 (0.49)	2.61 (0.62)	0.050 *
Palliative Care Volunteer Programs.	2.79 (0.46)	2.65 (0.56)	0.004 **
Networking. Integrated and person-centered end-of-life care.	2.72 (0.49)	2.61 (0.61)	0.070 *
Tools to care for and accompany people at the end of their lives.	2.82 (0.41)	2.74 (0.47)	0.067 *
Skills to practice compassion, emotional balance, empathy, and active listening in the professional and personal practice.	2.84 (0.42)	2.74 (0.49)	0.025 **
Research and evaluation in Palliative Care.	2.79 (0.45)	2.69 (0.54)	0.055 *
Management and organization of Palliative Care resources.	2.73 (0.50)	2.66 (0.55)	0.201
Compassionate community and network management.	2.73 (0.48)	2.60 (0.61)	0.034 **
Public policies for the development of Palliative Care.	2.67 (0.54)	2.57 (0.59)	0.065 *
Rights of end-of-life care.	2.84 (0.39)	2.72 (0.51)	0.013 **

Note: We show the mean values and standard deviations in brackets when the variable is numeric. We show the frequency and percentage in brackets when the variable is categorical. ***, **, and * represent statistically significant differences at 1%, 5%, and 10% between values of variables by the students’ course (Medicine, Nursing, Psychology) and by affiliation with the University (students, lecturers).

**Table 4 ijerph-17-05425-t004:** Gilbert’s Scale and the Life Engagement Test by students’ course and affiliation (n = 788).

Variables	Total Sample n = 788	Students n = 684			*p*-Value ^a^	Faculty Members n = 77	Administration Staff n = 27	*p*-Value ^b^
		Medicine n = 277	Nursing n = 27	Psychology n = 380				
**Gilbert’s Scale (2017): Compassion subscales**								
**Self-compassion**								
Engagement	32.19 (6.36)	41.52 (8.48)	41.07 (9.86)	42.22 (9.35)	0.021 **	42.25 (8.02)	42.30 (10.65)	0.926
Actions	41.94 (8.98)	31.22 (6.50)	32.89 (5.77)	32.74 (6.18)	0.007 **	32.04 (6.54)	34.15 (6.39)	0.143
Total	74.13 (13.63)	72.73 (13.41)	73.96 (14.07)	74.97 (13.69)	0.012 **	74.29 (13.11)	76.44 (15.68)	0.627
**Compassion for others**								
Engagement	41.77 (8.52)	41.53 (8.76)	42.30 (7.38)	41.98 (8.05)	0.399	41.34 (10.08)	41.96 (9.17)	0.894
Actions	31.15 (6.29)	29.83 (6.52)	31.30 (5.66)	31.96 (5.86)	0.002 **	31.29 (7.03)	32.70 (6.06)	0.320
Total	72.91 (13.44)	71.36 (14.28)	73.59 (11.90)	73.93 (12.27)	0.047 **	72.62 (15.75)	74.67 (14.02)	0.739
**Compassion from others**								
Engagement	35.49 (10.27)	35.66 (10.67)	36.63 (9.95)	35.38 (10.12)	0.875	35.99 (10.63)	32.74 (7.20)	0.185
Actions	25.92 (7.89)	25.63 (8.05)	27.07 (7.29)	26.03 (7.92)	0.453	25.55 (7.78)	27.30 (7.04)	0.726
Total	61.41 (17.01)	61.29 (17.72)	63.70 (16.42)	61.41 (16.84)	0.720	61.53 (17.34)	60.04 (11.53)	0.732
**Life Engagement Test**								
Total score	21.81 (3.74)	20.92 (3.78)	22.41 (3.92)	22.11 (3.68)	0.000 ***	23.06 (3.33)	22.74 (3.43)	0.003 **
Low score ≤20	307 (38.96)	148 (53.4)	7 (25.9)	128 (33.7)	0.000 ***	17 (22.1)	7 (25.9)	0.007 **
Medium score 21–25	336 (42.64)	90 (32.5)	14 (51.9)	180 (47.4)		39 (50.6)	13 (48.1)	
High score ≥26	145 (18.40)	39 (14.1)	6 (22.2)	72 (18.9)		21 (27.3)	7 (25.9)	

*Notes:* We show the mean values and standard deviations in brackets when the variable is numeric. We show the frequency and percentage in brackets when the variable is categorical. *** and ** represent statistically significant differences at 1% and 5% between values of variables by the students’ course ^a^ (Nursing, Medicine, Psychology) and by affiliation with the University ^b^ (students, faculty members, administration staff).

**Table 5 ijerph-17-05425-t005:** Gilbert’s Scale and the Life Engagement Test by gender among the students (n = 684).

Variables	Students		*p*-Value
	Women (n = 500)	Men (n = 184)	
**Gilbert’s Scale [34]: Compassion subscales**			
**Self-compassion**			
Engagement	42.10 (9.05)	41.32 (8.96)	0.329
Actions	32.32 (6.15)	31.62 (6.79)	0.326
Total	74.42 (13.33)	72.94 (14.35)	0.381
**Compassion for others**			
Engagement	42.12 (8.06)	40.97 (8.93)	0.174
Actions	31.49 (5.73)	29.92 (7.25)	0.046 **
Total	73.61 (12.40)	70.89 (14.84)	0.069 *
**Compassion from others**			
Engagement	35.77 (9.68)	34.92 (11.92)	0.736
Actions	26.27 (7.61)	24.94 (8.73)	0.15
Total	62.04 (16.23)	59.86 (19.45)	0.389
**Life Engagement Test**			
Total score	21.84 (3.72)	21.09 (3.88)	0.025 **
Low score ≤20	194 (38.8)	89 (48.4)	0.078 *
Medium score 21–25	216 (43.2)	68 (37.0)	
High score ≥26	90 (18.0)	27 (14.7)	

*Notes:* We show the mean values and standard deviations in brackets when the variable is numeric. **, and * represent statistically significant differences at 5% and 10% between values of variables by gender.

**Table 6 ijerph-17-05425-t006:** Correlation of the compassion subscales (n = 788).

	CEAS	1	2	3	4	5
**1**	**Self-compassion:** Engagement					
**2**	**Self-compassion:** Actions	**0.57 *****				
**3**	**Compassion for others:** Engagement	**0.64 *****	0.46 ***			
**4**	**Compassion for others:** Actions	0.49 ***	**0.59 *****	**0.61 *****		
**5**	**Compassion from others:** Engagement	0.42 ***	0.31 ***	0.49 ***	0.35 ***	
**6**	**Compassion from others:** Actions	0.38 ***	0.37 ***	0.40 ***	0.45 ***	**0.74 *****

*Notes:* CEAS = Compassionate Engagement and Actions Scales. *** The correlation is significant at the 0.01 level. The figures in bold indicate a strong correlation.

**Table 7 ijerph-17-05425-t007:** Correlation between the compassion subscales (Gilbert’s scale): self-compassion, compassion for others, compassion from others; and the Life Engagement Test among the students (n = 684).

		1	2	3	4	5	6
**1**	**Self-compassion:** Engagement						
**2**	**Self-compassion:** Actions	**0.57 *****					
**3**	**Compassion for others:** Engagement	**0.64 *****	0.46 ***				
**4**	**Compassion for others:** Actions	0.49 ***	**0.59 *****	**0.61 *****			
**5**	**Compassion from others:** Engagement	0.42 ***	0.31 ***	0.49 ***	0.35 ***		
**6**	**Compassion from others:** Actions	0.38 ***	0.37 ***	0.40 ***	0.45 ***	**0.74 *****	
**7**	**Life Engagement Test**	0.29 ***	0.44 ***	0.21 ***	0.32 ***	0.14 ***	0.23 ***

*** The correlation is significant at the 0.01 level. The figures in bold indicate a strong correlation.

**Table 8 ijerph-17-05425-t008:** Willingness to dedicate themselves professionally to Palliative Care among students and lecturers (n = 536). Univariate and multivariate regression models.

Willingness to Dedicate Themselves Professionally to Palliative Care [yes]	OR_CRUDE_ (95% IC)	OR_ADJUSTED_ (95% IC)
Age [centered on 22 years old]	1.01 (0.99–1.03)	1.03 (1.00–1.05) **
Gender [Female]	2.14 (1.47–3.16) ***	2.06 (1.39–3.08) ***
Affiliation with the University [Lecturer]	0.73 (0.44–1.19)	0.47 (0.24–0.91) **
Self-compassion scale	1.03 (1.02–1.04) ***	1.01 (0.99–1.03)
Compassion from others	1.11 (1.01–1.03) **	1.00 (0.99–1-01)
Compassion for others	1.04 (1.02–1.05) ***	1.02 (1.00–1.04) **
Feeling trained to accompany a person in the stages of their disease [yes]	1.74 (1.20–2.53) **	1.42 (0.95–2.11) *
Life Engagement Test	1.06 (1.01–1.11) **	1.01 (0.96–1.07)

*Notes*: ***, **, and * represent statistically significant differences at 1%, 5%, and 10%.

**Table 9 ijerph-17-05425-t009:** Willingness to dedicate themselves professionally to Palliative Care among Medicine, Nursing and Psychology students (n = 459). Univariate and multivariate regression models.

Willingness to Dedicate Themselves Professionally to Palliative Care [yes]	OR_CRUDE_ (95% IC)	OR_ADJUSTED_ (95% IC)
Age [centered on 22 years old]	1.03 (0.99–1.05) *	0.99 (0.96–1.03)
Gender [Female]	2.13 (1.41–3.23) ***	1.81 (1.17–2.82) **
Affiliation with the University [Medicine student]		
Nursing student	2.24 (1.01–5.14) **	1.77 (0.77–4.19)
Psychology student	2.44 (1.64–3.67) ***	2.11 (1.31–3.42) **
Self-compassion scale	1.03 (1.02–1.05) ***	1.01 (0.98–1.03)
Compassion from others	1.02 (1.01–1.03) **	1.00 (0.99–1.02)
Compassion for others	1.04 (1.02–1.05) ***	1.02 (1.00–1.05) **
Feeling trained to accompany a person in the stages of their disease [yes]	1.60 (1.07–2.38) **	1.32 (0.86–2.03)
Life Engagement Test	1.08 (1.03–1.13) **	1.02 (0.96–1.08)

*Notes:* ***, **, and * represent statistically significant differences at 1%, 5%, and 10%.

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
