# Peer review of "Compassionate Engagement and Action in the Education for Health Care Professions: A Cross-Sectional Study at an Ecuadorian University"

_ijerph, 2020, doi:10.3390/ijerph17155425_

Round 1
Reviewer 1 Report
The paper describes a high-quality cross-sectional study that explains clearly the methods used and the study's findings in relation to the study purposes set in the end of the introduction and discusses the contribution and limitations of those findings
Notes:
- In lines 81-87 the authors present background information about the concept which is important as they specifically measure it in their study. In lines: 124-126 the authors noted that their measures also measures “self-compassion, compassion for others, and compassion from others” so it would be worthwhile adding a bit information about the different types compassion that they mention and why it is important to address all of them.
- In lines 120-125 the authors describe the different measures that they use. Two of their measures were a previously validated questionnaire but it would be helpful to provide some information on how the items for the other blocks were formed (lines: 121-123). Those included a) level of knowledge and sensitivities and b) training needs of the professors and students. Were those constructed based on previous literature or were there specific content guidelines followed?
- One suggestion is to move the larger table (table 3) in an attachment so the reader can read the content of that section more easily, while also being able to have the table available separately for their reference.
- In table 9 the authors signpost Affiliation with the University [Medicine student]:Nursing student and Psychology student, but in table 8 that they signpost Affiliation with university they only report [lecturer]. In lines 336-337 they note that “lecturers in comparison to students were found to present lower 336 willingness to dedicate themselves professionally to palliative care” thus should table 7 also report model findings for [students]?
- One discussion point may be that differences found through the regression analysis (between lecturers and students and between nursing and psychology students) may mean that future training needs may also differ for these groups as they move on in different career trajectories, thus future research should seek to see whether difference at baseline seem to be persistent or not.
Author Response
The paper describes a high-quality cross-sectional study that explains clearly the methods used and the study's findings in relation to the study purposes set in the end of the introduction and discusses the contribution and limitations of those findings
Notes:
- In lines 81-87 the authors present background information about the concept which is important as they specifically measure it in their study. In lines: 124-126 the authors noted that their measures also measures “self-compassion, compassion for others, and compassion from others” so it would be worthwhile adding a bit information about the different types compassion that they mention and why it is important to address all of them.
Response: We used these terms based on the Gilbert´s work that was referenced. Nevertheless, following this suggestion, the definitions were included as well. Please, see lines 161-165.
- In lines 120-125 the authors describe the different measures that they use. Two of their measures were a previously validated questionnaire but it would be helpful to provide some information on how the items for the other blocks were formed (lines: 121-123). Those included a) level of knowledge and sensitivities and b) training needs of the professors and students. Were those constructed based on previous literature or were there specific content guidelines followed?
Response: Thank you for this observation, the first three blocks were based on previous literature and the last two blocks were based on validated questionnaires. This idea was introduced into the text. Please see lines 158-159.
- One suggestion is to move the larger table (table 3) in an attachment so the reader can read the content of that section more easily, while also being able to have the table available separately for their reference.
Response: Thank you for this suggestion that was taken into account. Now, the table can be consulted in an attachment section.
- In table 9 the authors signpost Affiliation with the University [Medicine student]: Nursing student and Psychology student, but in table 8 that they signpost Affiliation with university they only report [lecturer]. In lines 336-337 they note that “lecturers in comparison to students were found to present lower 336 willingness to dedicate themselves professionally to palliative care” thus should table 7 also report model findings for [students]?
Response: Thank you very much for this comment. In table 9, students of different academic degrees are compared, having three population groups: medical, nursing and psychology students. However, in Table 8 only students in general are compared with teachers. Being two comparison groups, we only indicate the reference category next to the affiliation, understanding that in this case he is a teacher and that value is compared with the students. We have changed the title for a better understanding. As for the comment present on lines 336-337, it is referred to table 8, therefore, the doubt about the inclusion of students would be resolved. We have corrected this error in the text.
- One discussion point may be that differences found through the regression analysis (between lecturers and students and between nursing and psychology students) may mean that future training needs may also differ for these groups as they move on in different career trajectories, thus future research should seek to see whether difference at baseline seem to be persistent or not.
Response: We consider this observation very useful and was included in the discussion. Please, see lines 458-459.
Reviewer 2 Report
This is a well-structured, well-presented, and generally well-written article. While it is not ground-breaking in its approach and content, it is an important step forward in increasing awareness about the importance of compassion in training healthcare professionals in Ecuador. It also offers a good review of the literature on the needs for compassion in healthcare, and why it is important to train students in Nursing, Medicine, and Psychology in compassion (e.g., the expectations from patients and the important beneficial effects). It also offers an important perspective on gender differences and also the lack of compassion training in Medicine, which needs to be addressed, not just in Ecuador but around the world.
While the manuscript is generally well-structured and well-written, the Discussion section seems a bit disjointed, and the Conclusion seems too short. I would suggest moving some parts of the Discussion to the Introduction on previous literature (e.g., some or all of the paragraphs in Lines 456-462, Lines 477-491), as it seems to be going back to the previous research. In addition, some of the Discussion could be moved to the Conclusion, such as Lines 463-475.
Another minor issue is that in the current PDF version, the line spacing among the paragraphs seems to be inconsistent.
Author Response
This is a well-structured, well-presented, and generally well-written article. While it is not ground-breaking in its approach and content, it is an important step forward in increasing awareness about the importance of compassion in training healthcare professionals in Ecuador. It also offers a good review of the literature on the needs for compassion in healthcare, and why it is important to train students in Nursing, Medicine, and Psychology in compassion (e.g., the expectations from patients and the important beneficial effects). It also offers an important perspective on gender differences and also the lack of compassion training in Medicine, which needs to be addressed, not just in Ecuador but around the world.
Response: Thank you for your kind words.
While the manuscript is generally well-structured and well-written, the Discussion section seems a bit disjointed, and the Conclusion seems too short. I would suggest moving some parts of the Discussion to the Introduction on previous literature (e.g., some or all of the paragraphs in Lines 456-462, Lines 477-491), as it seems to be going back to the previous research.
Response: We addressed this recommendation and this part was restructured following your suggestion. Please, see lines 101-122.
In addition, some of the Discussion could be moved to the Conclusion, such as Lines 463-475.
Response: Thank you for this suggestion that was taken into account. Now, you may find that lines in conclusion section. References were deleted, and re numbered the discussion section. Please, see lines 584-594.
Another minor issue is that in the current PDF version, the line spacing among the paragraphs seems to be inconsistent.
Response: This observation was considered and all paragraphs were reviewed.
Reviewer 3 Report
The authors present a timely and important topic. This manuscript presents a cross-sectional study to analyze the compassion attitudes in medical health professionals. They followed STROBE guidelines for cross-sectional studies. Overall this was a nicely written paper. I have read carefully and found that this study is very carefully created and developed. Although this study has scientific interest, some important aspects should be reviewed by the authors. I hope that my opinions will help shape your research article more precise and interesting. The followings are my comments:
- Method: I suggest detailing if the questionnaire used [Gilbert, P.; Catarino, F.; Duarte, C.; Matos, M.; Kolts, R.; Stubbs, J.; Ceresatto, L.; Duarte, J.; 598 Pinto-Gouveia, J.; Basran, J. The development of compassionate engagement and action scales for self and others. J Compassionate Health Care 2017, 4(1), 4.] has been validated in Ecuador. If not, if the authors can include some data about reliability and validity.
- Results: I suggest explaining why only 27 nursing students have answered the questionnaire. Compassion is very important in nursing.
- Discussion: Future research should be addressed for this very important topic.
Author Response
The authors present a timely and important topic. This manuscript presents a cross-sectional study to analyze the compassion attitudes in medical health professionals. They followed STROBE guidelines for cross-sectional studies. Overall this was a nicely written paper. I have read carefully and found that this study is very carefully created and developed. Although this study has scientific interest, some important aspects should be reviewed by the authors. I hope that my opinions will help shape your research article more precise and interesting. The followings are my comments:
- Method: I suggest detailing if the questionnaire used [Gilbert, P.; Catarino, F.; Duarte, C.; Matos, M.; Kolts, R.; Stubbs, J.; Ceresatto, L.; Duarte, J.; 598 Pinto-Gouveia, J.; Basran, J. The development of compassionate engagement and action scales for self and others. J Compassionate Health Care 2017, 4(1), 4.] has been validated in Ecuador. If not, if the authors can include some data about reliability and validity.
- Results: I suggest explaining why only 27 nursing students have answered the questionnaire. Compassion is very important in nursing.
Response: We included the reason of this in methodology section. We agree that compassion is very important in nursing degree but this career started two years ago in this university and there are few students so far.
Discussion: Future research should be addressed for this very important topic.
Response: This idea was included. Please, see line 517.
Reviewer 4 Report
Thank you very much for your scientific contributions. The manuscript develops a current theme. Below are a number of observations in case they might be useful.
TITLE
Adequate.
SUMMARY
It is recommended that some aspects be modified. For example, the word Table should not appear since the summary does not refer to a specific table. This information should be placed later, throughout the manuscript and as close as possible to the table in question.
It would be interesting to have more specific data on the sample. In the title, the location of the University is shown. It should also appear in the abstract. It would also be useful to know the percentage of men or women in relation to the total number of participants.
Using the STROBE guide is a great idea.
KEY WORDS
Check the magazine guidelines to ensure the maximum number of keywords.
Use at least one keyword to describe the research design.
It is recommended that you put the words in alphabetical order if no other order is implied.
INTRODUCTION
Adequate. A general objective and several specific objectives are set out at the end. The use of hypotheses based on these specific objectives is recommended for discussion.
METHODOLOGY
Generally speaking, it is adequate. There are some details that should be made explicit as the Cronbach Alpha of the instruments used.
RESULTS
Check out the blank page fragments that are left in several different places in the manuscript.
DISCUSSION
During the introduction, 23 references will be used. However, the bibliographic references used in the discussion section do not link to the theoretical framework that has served as the basis for the study. The same references are not used. This aspect needs to be improved. It is necessary that a large part of the bibliographic references used in the theoretical foundation also form part of the discussion, some of which may be new, but not the vast majority.
It would be convenient that the order of writing in the discussion be based on the hypotheses to which I have alluded in the review of the introduction.
The manuscript develops the applicability of the study, the limitations and the future lines of research, which is positive.
CONCLUSIONS
Adequate. It correctly summarizes the main conclusions of the study.
REFERENCES
Check the format. Overall it's fine. Only some details have been detected to be modified, such as the absence of spaces between the year of publication and other data.
In summary, it is an interesting work with strong points such as the updated bibliography and the structure used. The discussion and the introduction should be modified in order to link the ideas and to be able to discuss them in depth. Thank you very much for your work and for your attention.
Author Response
Thank you very much for your scientific contributions. The manuscript develops a current theme. Below are a number of observations in case they might be useful.
TITLE
Adequate.
Response: Thank you.
SUMMARY
It is recommended that some aspects be modified. For example, the word Table should not appear since the summary does not refer to a specific table. This information should be placed later, throughout the manuscript and as close as possible to the table in question.
Response: We agree with you, actually, this was a mistake. Thank you for the observation.
It would be interesting to have more specific data on the sample. In the title, the location of the University is shown. It should also appear in the abstract.
Response: This suggestion was included now.
It would also be useful to know the percentage of men or women in relation to the total number of participants.
Response: This data is included in table 1: A total of 73.22% are female students and 26,78% are male students.
Using the STROBE guide is a great idea.
Response: Thank you.
KEY WORDS
Check the magazine guidelines to ensure the maximum number of keywords.
Use at least one keyword to describe the research design.
Response: Done. We checked instructions for authors information and the maximum of keywords are ten. We included seven.
It is recommended that you put the words in alphabetical order if no other order is implied.
Response: Done
INTRODUCTION
Adequate. A general objective and several specific objectives are set out at the end. The use of hypotheses based on these specific objectives is recommended for discussion.
Response: Thank you.
METHODOLOGY
Generally speaking, it is adequate. There are some details that should be made explicit as the Cronbach Alpha of the instruments used.
Response: Thank you very much for this comment. We have included some details of the instruments used in section Material and Methods – 2.3. Measures.
RESULTS
Check out the blank page fragments that are left in several different places in the manuscript.
Response: This is due to the size of the tables. We hope to solve this when the paper is ready to publish. Editor assistant usually helps with this. Thank you for the observation.
DISCUSSION
During the introduction, 23 references will be used. However, the bibliographic references used in the discussion section do not link to the theoretical framework that has served as the basis for the study. The same references are not used. This aspect needs to be improved. It is necessary that a large part of the bibliographic references used in the theoretical foundation also form part of the discussion, some of which may be new, but not the vast majority. It would be convenient that the order of writing in the discussion be based on the hypotheses to which I have alluded in the review of the introduction.
Response: Our purpose was to introduce compassion in the introduction section in a theoretical level and later, in discussion, our intention was to compare our findings with others authors’ findings. However, both sections were modified in order to address this suggestion.
The research team did not consider to stablish hypothesis because this is a cross-sectional design that pretended to explore what happened regarding this topic.
The manuscript develops the applicability of the study, the limitations and the future lines of research, which is positive.
Response: Thank you.
CONCLUSIONS
Adequate. It correctly summarizes the main conclusions of the study.
Response: Thank you.
REFERENCES
Check the format. Overall it's fine. Only some details have been detected to be modified, such as the absence of spaces between the year of publication and other data.
Response: Thanks for this observation. All references were reviewed.
In summary, it is an interesting work with strong points such as the updated bibliography and the structure used. The discussion and the introduction should be modified in order to link the ideas and to be able to discuss them in depth. Thank you very much for your work and for your attention.
Response: Thank you for this observation. We linked ideas between two sections. We hope you find this proper now.
Round 2
Reviewer 4 Report
Thanks for all the work done to improve the manuscript. The effort and improvements made are evident. The only aspect that, in my opinion, should be modified, is the one related to DISCUSSION. It would be desirable that more references used previously in the introduction be used in the discussion.
Thank you for your attention.
Author Response
Thanks for all the work done to improve the manuscript. The effort and improvements made are evident. The only aspect that, in my opinion, should be modified, is the one related to DISCUSSION. It would be desirable that more references used previously in the introduction be used in the discussion.
Thank you for your attention.
Response: Thank you for this suggestion. Following this, we introduced some references from introduction section to discussion section supporting the ideas provided in this part. We hope you find these changes suitable.